# Dense Matrix Multiplication Algorithms and Performance Evaluation of HPCC in 81 Nodes IBM Power 8 Architecture

**Eduardo Patricio Estévez Ruiz** [1], **Giovanny Eduardo Caluña Chicaiza** [2], **Fabian Rodolfo Jiménez Patiño** [3], **Joaquín Cayetano López Lago** [1] **and Saravana Prakash Thirumuruganandham** [4,*]

1   Grupo de Polímeros, Departamento de Física y Ciencias de la Tierra, Escuela Universitaria Politécnica, Universidade da Coruña, 15471 Ferrol, Spain; eduardo45pr@gmail.com (E.P.E.R.); joaquin.lopez@udc.es (J.C.L.L.)
2   School of Mathematical Sciences and Information Technology, Yachay Tech University, Urcuquí 100119, Ecuador; giovanny.caluna@yachaytech.edu.ec
3   Department of Technological Operations, Alluriam Inc. Co., Orlando, FL 32801, USA; fabian.jimenez@alluriam.com
4   Centro de Investigación en Mecatrónica y Sistemas Interactivos—MIST del Instituto de Investigación, Desarrollo e Innovación, Universidad Tecnológica Indoamérica (UTI), Ambato 180103, Ecuador
*   Correspondence: saravprak@googlemail.com or saravanaprakash@uti.edu.ec; Tel.: +593-994-76-5516

**Abstract:** Optimizing HPC systems based on performance factors and bottlenecks is essential for designing an HPC infrastructure with the best characteristics and at a reasonable cost. Such insight can only be achieved through a detailed analysis of existing HPC systems and the execution of their workloads. The "Quinde I" is the only and most powerful supercomputer in Ecuador and is currently listed third on the South America. It was built with the IBM Power 8 servers. In this work, we measured its performance using different parameters from High-Performance Computing (HPC) to compare it with theoretical values and values obtained from tests on similar models. To measure its performance, we compiled and ran different benchmarks with the specific optimization flags for Power 8 to get the maximum performance with the current configuration in the hardware installed by the vendor. The inputs of the benchmarks were varied to analyze their impact on the system performance. In addition, we compile and compare the performance of two algorithms for dense matrix multiplication SRUMMA and DGEMM.

**Keywords:** supercomputer; performance; benchmark; IBMPower8; HPC; Cluster; DGEMM; SRUMMA; Parallel Computing



## 1. Introduction

Today's computational methods such as modeling and simulations are the core tools for finding solutions to various biological to engineering problems that can only be solved by an HPC platform. Performance evaluation and analysis have been a core topic in HPC research. Formally, a benchmark is a program or set of programs executed on a single machine or cluster to obtain the maximum performance of a given function under certain conditions and to compare the performance results with the measured values of similar machines or the theoretically expected values. High-Performance Linpack (HPL) benchmark, used in the TOP500 list as a long-established standard for measuring computational performance, has been challenged in recent years [1,2] . The LINPACK (HPL) benchmark was the defacto metric for ranking HPC systems, measuring the sustained floating point rate (GFLOPs/s) for solving a dense system of linear equations using double precision floating point arithmetic. The use of HPCC in performance modeling and prediction has already been investigated in the following work: Refs. [1–4]. In particular, Pfeiffer et al. [5] applied linear regression to adjust the execution time of scientific applications based on the speeds and latencies of the HPCC cores. Their models for the HPL and G-FFTE benchmarks in HPCC also derive the functional dependencies of problem size and core count from

complexity analysis. On the other hand, Chen et al. [6] demonstrated a statistical approach that combines Variable Clustering (VarCluster) and Principal Component Analysis (PCA) to rigorously compare the adequacy and representativeness of a benchmark for real-world HPC workloads. It was investigated which metrics are the best predictors of scientific application performance, and predicting the bandwidth of strided accesses to main memory or the bandwidth of random accesses to the L1 cache provides more accurate predictions than floating-point-operations-per-second (flops), on which the Top 500 ranking is based. It also defined how a combination of application and machine characteristics can be used to compute improved workload-independent rankings. A notable example is that Sayeed et al. [7] recommend that small benchmarks cannot anticipate the behavior of real-world HPC applications. They discuss important issues, challenges, tools, and metrics in HPC application performance evaluation. They then evaluate the performance of four application benchmarks on three different parallel architectures, estimating runtime, inter-process communication overhead, I/O characteristics, and memory requirements. Because they measure these metrics on a variety of implementation processes, the results differ from one execution to another. From their results, it appears that on different numbers of execution processes, different platforms perform better or worse, which can significantly benefit the test at a particular scale of the experiment. Marjanovic et al. [8] attempted a performance model for the HPCG benchmark, they analyzed the impact of the input dataset for three representative benchmarks: HPL, HPCG, and High-Performance Geometric Multigrid (HPGMG), and performed a node-level analysis on six specific HPC platforms, performing a scale-out analysis on one of the platforms. Their results show that examining multiple problem sizes gives a more complete picture of basic system performance than a single number representing the best performance. In recent years, institutions running HPC applications have focused on areas ranging from molecular dynamics to an-imation to weather forecasting. One approach to achieve higher computational performance has been to add graphics processing units (GPUs) that can serve the complex, demanding computational needs, such as the NVIDIA Tesla K80GPU accelerators, to the dedicated server hardware that can offload or supplement the workload on the CPU. There are not many studies that deal with benchmarking techniques and how to estimate the results of HPC systems and applications in relation to Tesla K80 and identification analysis of previous versions. For this reason, it is necessary for dedicated HPC systems such as the supercomputer "Quinde I" to know its full potential and optimize it under real scenarios, which is usually difficult to achieve with small tests [9–11], and this is where the need to apply benchmarking becomes apparent. In computer science, benchmarking is a technique to measure the performance of a system or one of its components. A benchmark is a well-defined and simple program [12]. Based on the results obtained when running a benchmark, the maximum real-world performance can be measured and compared. In HPC, benchmarking is used to get a better understanding of the weaknesses and strengths of the system [13]. It is necessary to have a good understanding of how an application performs (or scales) on different architectures in order to:

- Estimate the average runtime for a single job.
- Reduce the waiting time to start a job.
- To set a proper size for the jobs.
- Improve the customization process on a given application for accessing large centers such as Edinburgh Parallel Computing Center (EPCC) [14].

Although there is much debate about how relevant the HPL benchmark is to the industry, it remains an excellent "burn-in" test for very complex HPC systems. HPL is a good tool to validate a system: it works as a control and stresses the system more than typical applications would.With this in mind, the HPL benchmark is used in this work to validate the system. The goal of this work is to present a performance analysis combined with an architectural analysis of CPUs and GPUs on our K80 machine, providing an understanding that both CPUs and GPUs should be considered as complementary hybrid solutions for many scientific applications.

- We demonstrate HPL performance on our HPC "Quinde I", which is equipped with high technology with the following features: a cluster with IBM Power 8 processors, 1760 cores, RAM 11 TB, parallel memory 350 TB, NVIDIA K80 Tesla GPU and InfiniBand technology. Based on the official linpack results, "Quinde I" achieves 231.9 teraflops, we discuss two different linpacks to measure the performance of CPUs and GPUs independently.
- We show how appropriate microarchitectural benchmarking can be used to reveal the cache performance characteristics of modern Tesla K80 by NVIDIA processors between CPU and GPU's, specifically parameterizing the performance of possible multiple layers of cache present on and off the processor, we detailed the Cache-Bench which contains eight different sub-benchmarks, related to raw bandwidth in floating point workloads and compared the performance characteristics of CPU and GPU's.
- We show how the optimization problem can be parallelized on both traditional CPU based systems and GPU and compare their performance
- We show the CUDA implementation of the assembly step in NVIDIA 80, which is straightforward.
- For DGEMM, we show the effect of varying core and detail problem size in terms of node performance after testing and speed up of each node of "Quinde I".
- For Shared and Remote-memory based Universal Matrix Multiplication Algorithm (SRUMMA) [15], a parallel algorithm implementing dense matrix multiplication with algorithmic efficiency, experimental results on clusters (16-way IBM SP and 2-way Linux/Xeon nodes) and shared-memory systems confirm SGI Altix, Cray X1 with its partitioned shared memory hardware SUMMA [16] that with reference to previous studies, SRUMMA demonstrates consistent performance advantages over the pdgemm routine from the ScaLAPACK/PBBLAS suite [17,18], the leading implementation of the parallel matrix multiplication algorithms used today. Considering such factors and the impact of SRUMMA on various such clusters, we evaluated the performance of SRUMMA on the IBM power 8 architecture.

Therest of this paper is organized as follows: Section 2 describes the experimental platform and its computing system architectures and describes the benchmark methods used; Section 3 describes the results ofthe HPL, STREAM, Babel, Cache benchmark suite along with the individual performance of DGEMM and the theoretical performance, while Section 4 describes the SRUMMA vs. DGEMM algorithms, and finally Section 5 contains the conclusions of the study.

## 2. Materials and Methods

### 2.1. Experimental Platform

We performed the experiments onthe supercomputer "Quinde I", which consisted of 84 compute nodes, two login nodes and two management nodes interconnected by a high-speed InfiniBand EDR network at 100 Gb/s. It had a theoretical performance (RPEAK) of 488.9 TFLOPs and a maximum performance (RMAX) of 231.9 LINPACK TFLOPs.

It was managed in the form of a cluster by the SCF administration node as shown in Figure 1, and received the jobs to be processed from the administration node of the LSF queues, the work orders were sent to the nodes through the login node which had access to the users; all the processed information was stored in a parallel GPFS file system through the Infiniband high performance network; in addition, all the issues related to the performance of the devices were monitored by independent networks, which avoided the junk traffic. Quinde-I consisted of servers (nodes) of IBM model 822LC, with Power8 processors with MIND technology (NUMA) with 128 GB of RAM and 1000 GB local hard disk, in addition each node had two graphics processing cards of NVIDIA brand model K80 Tesla, and four network interfaces; each of them with a dedicated purpose. Each computing node has two power eight processors model 8335GTA, with 10 cores each, where each core in turn hasd the capacity to process eight threads, which means that each node had 160 threads (see Figure 2).

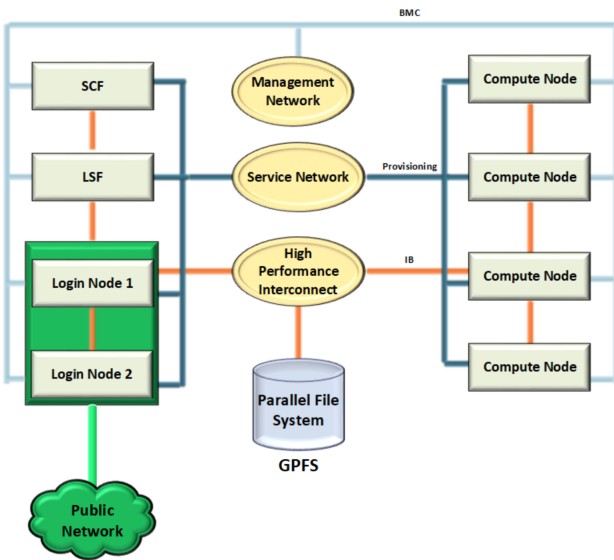

**Figure 1.** General logic diagram of Quinde I.

| | |
|---|---|
| Architecture | ppc64le |
| Byte Order: | Little Endian |
| CPU(s): | 160 |
| On-line CPU(s) | $0-159$ |
| Thread(s) per core: | 8 |
| Core(s) per socket: | 10 |
| Socket(s): | 2 |
| NUMA node(s): | 2 |
| Model: | $8335-GTA$ |
| L1d cache: | 64k |
| L1i cache: | 32k |
| L2 cache: | 512k |
| L3 cache: | 8192k |
| NUMA node0 CPU(s): | $0-79$ |
| NUMA node8 CPUS(s): | $80-159$ |

**Figure 2.** Characteristics of the cores of Supercomputer Quinde I node.

## 2.2. High Performance Linpack (HPL)

One of the most important and widely used tests on HPC is the HPL, which measures the floating-point computational performance of the system. This test is used in the TOP 500, a ranking of the 500 supercomputers with the highest performance in the world. To obtain the HPL measure, we ran the benchmark to solve a dense **n** by **n** system of linear equations:

$$\mathbf{Ax = b} \tag{1}$$

To run the HPL, it was necessary to set various values in a main file. The input file contained various parameters that HPL needed to run the test. These parameters have a great impact on the test performance, so they should be chosen carefully. One of the most important parameters is **N**. It is the dimension of the coefficient matrix generated to solve the linear system. To select a **N** and find out the best performance for a system, the memory system must be considered. If the selected problem size **N** is too large, swapping will occur and the performance will decrease. Another parameter to consider when running the test is **NB**. HPL uses the block size **NB** for both data distribution and computational granularity, noting that a small **NB** improves load balancing, but on the other hand, a very small **NB** can limit computational performance since there is almost no data reuse at the highest level of the memory hierarchy. Finally, we have the values P and Q that define the size of the grid. The product of these values must be equal, close to the number of processors in

the cluster, but no larger. According to [19], a smaller P X Q matrix dimension gives the best improvement.

### 2.3. Stream Benchmark

The STREAM benchmark is a simple benchmark program that measures sustainable memory bandwidth (in MB/s) and corresponding computation rate for large datasets [20]. The stream benchmark becomes relevant because most systems create a bottleneck because they are limited by memory bandwidth rather than processor speed.

### 2.4. DGEMM Benchmark

DGEMM [21] measures the floating point rate of execution of a real matrix–matrix multiplication with double precision. The code is designed to measure the sustained floating-point computation rate of a single node.

### 2.5. Babel Stream

Babel Stream [22] measures memory transfer rates to/from global device memory on GPUs. This benchmark is similar and based on the STREAM benchmark [20] for CPUs mentioned above. There are several implementations of this benchmark in a variety of programming models. Currently implemented are:

- OpenCL
- CUDA
- OpenACC
- OpenMP 3 and 4.5
- Kokkos
- RAJA
- SYCL

For this work, the version CUDA was used.

### 2.6. Cache Bench

Cache-Bench is one of the three tests of LLCBenchmark [23] designed to evaluate the performance of the memory hierarchy of computer systems. Its specific focus is on parameterizing the performance of possibly multiple levels of cache present on and off the processor. By performance, we mean raw bandwidth for floating-point workloads. Cache-Bench includes eight different sub-benchmarks. Each of them performs repeated accesses to data items in different vector lengths. The test takes the time for each vector length over a number of iterations. The accessed data in bytes are the product of iterations and vector length. Then the data are divided by the total time to calculate the bandwidth.

1. **Cache Read:** This benchmark is designed to give us the read bandwidth for different vector lengths in a compiler optimized loop.
2. **Cache Write:** This benchmark is designed to give us the write bandwidth for different vector lengths in a compiler optimized loop.
3. **Cache Read/Modify/Write:** This benchmark is designed to provide us with read/modify/write bandwidth for varying vector lengths in a compiler optimized loop. This benchmark generates twice as much memory traffic, as each data item must be first read from memory/cache to register and then back to cache.
4. **Hand tuned Cache Read:** This benchmark is a modification of Cache Read. The modifications reflect what a minimally good compiler should do for these simple loops.
5. **Hand tuned Cache Write:** This benchmark is a modification of Cache Write. The modifications reflect what a minimally good compiler should do on these simple loops.
6. **Hand tuned Cache Read/Modify/Write:** This benchmark is a modification of Cache Read/Modify/Write. The modifications reflect what a minimally good compiler should do with these simple loops.

7. **memset() from the C library:** The C library has the memset() function to initialize memory areas. With this benchmark, we can compare the performance of the two formulations for writing the memory.
8. **memcpy() from the C library:** The C library has the memcpy() function for copying memory areas. With this benchmark, we can compare the performance of the two versions of memory read/modify/write with this version.

## 2.7. Effective Bandwidth $b_{eff}$

The effective bandwidth $b_{eff}$ measures the cumulative bandwidth of the communication network of a parallel and/or distributed computing system [24]. Different message sizes, communication patterns and methods are used. The Algorithm 1 uses an average to account for the fact that in real-world applications, short and long messages result in different bandwidth values (shown in Figures 3–5 and Tables 1–3).

---

**Algorithm 1** The algorithm of $b_{eff}$

---

$b_{eff}$ = logavg ( logavgcartesian pattern (sumL (maxmthd (maxrep ( b(cartes.pat.,L,mthd,rep) )))/21 ),
logavg-random pattern(sumL (maxmthd (maxrep ( b(random pat.,L,mthd,rep) )))/21 ))

---

Effective Bandwith Benchmark ($b_{eff}$) Version 3.5
High-Performance Computing
Mon March 9 22:42:00 2020 on Linux it01-r10-cn-36.yachay.ep 3.10.0-327.el7.ppc64le Number 1 SMP Thu Oct 29 17:31:13 EDT 2015 ppc64le
**b$_{eff}$** = 17,556.798 MB/s = 877.840 * 20 PEs with 128 MB/PE

**Table 1.** Bandwidth with 20 core.

| | number of processors | $b_{eff}$ | Lmax | $b_{eff}$ at Lmax rings and random | $b_{eff}$ at Lmax rings only |
| --- | --- | --- | --- | --- | --- |
| | | MByte/s | | MByte/s | MByte/s |
| accumulated per process | 20 | 17,557 878 | 1 MB 2551 | 51,010 2501 | 50,013 |
| | Latency rings and random microsec | Latency rings only microsec | Latency ping-pong microsec | ping-pong bandwidth MByte/s | |
| accumulated | 1.970 | 1.808 | 0.782 | 14,893 | |

In the table, Ping-Pong result (only the processes with rank 0 and 1 in MPI_COMM_WORLD were used): Latency: 0.782 microsec per message; Bandwidth: 14,892.569 MB/s (with MB/s = $10^6$ byte/s).

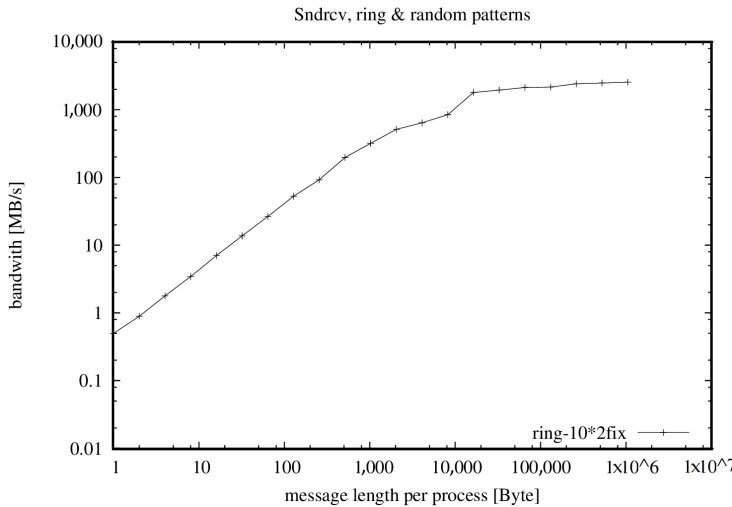

**Figure 3.** Bandwidth with 20 cores, one per node.

Sun March 15 23:31:05 2020 on Linux it01-r14-cn-63.yachay.e3.10.0-327.el7.ppc64le #1 SMP
Thu Oct 29 17:31:13 EDT 2015 ppc64le
$\mathbf{b_{eff}}$ = 28,590.215 MB/s = 893.444 $*$ 32 PEs with 128 MB/PE

**Table 2.** Bandwidth with 32 core.

|  | number of processors | $b_{eff}$ | Lmax | $b_{eff}$ at Lmax rings and random | $b_{eff}$ at Lmax rings only |
|---|---|---|---|---|---|
|  |  | MByte/s |  | MByte/s | MByte/s |
| accumulated per process | 32 | 28,590 893 | 1 MB 2428 | 77,698 2814 | 90,036 |
|  | Latency rings and random microsec | Latency rings only microsec | Latency ping-pong microsec | ping-pong bandwidth MByte/s |  |
| accumulated | 1.857 | 1.681 | 0.788 | 18,475 |  |

In the table, Ping-Pong result (only the processes with rank 0 and 1 in MPI_COMM_WORLD were used): Latency: 0.788 microsec per message; Bandwidth: 18,475.306 MB/s (with MB/s = $10^6$ byte/s).

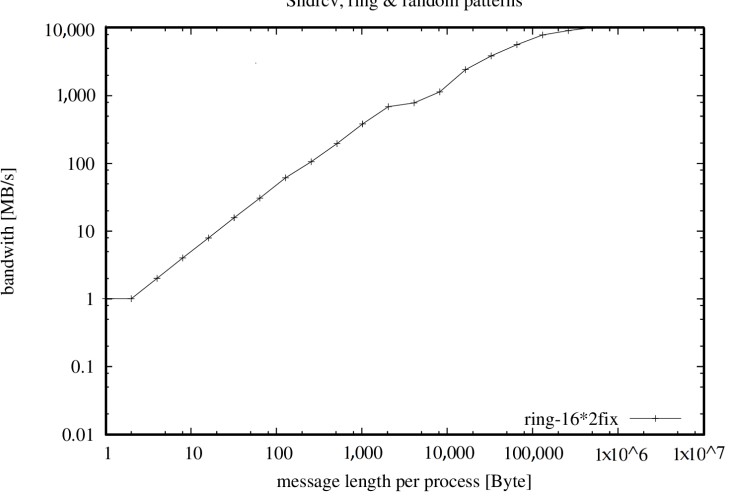

**Figure 4.** Bandwidth with 32 cores, one per node.

Sun March 15 23:34:36 2020 on Linux it01-r10-cn-39.yachay.ep 3.10.0-327.el7.ppc64le #1 SMP Thu Oct 29 17:31:13 EDT 2015 ppc64le
$\mathbf{b_{eff}}$ = 62,367.350 MB/s = 974.490 $*$ 64 PEs with 128 MB/PE

**Table 3.** Bandwidth with 64 core.

| | number of processors | $b_{eff}$ | Lmax | $b_{eff}$ at Lmax rings and random | $b_{eff}$ at Lmax rings only |
| --- | --- | --- | --- | --- | --- |
| | | MByte/s | | MByte/s | MByte/s |
| accumulated per process | 64 | 62,367 974 | 1 MB 2828 | 180,979 3916 | 250,604 |
| | Latency rings and random microsec | Latency rings only microsec | Latency ping-pong microsec | ping-pong bandwidth MByte/s | |
| accumulated | 1.918 | 1.699 | 0.787 | 20,682 | |

In the table, Ping-Pong result (only the processes with rank 0 and 1 in MPI_COMM_WORLD were used): Latency: 0.787 microsec per message; Bandwidth: 20,682.090 MB/s (with MB/s = $10^6$ byte/s).

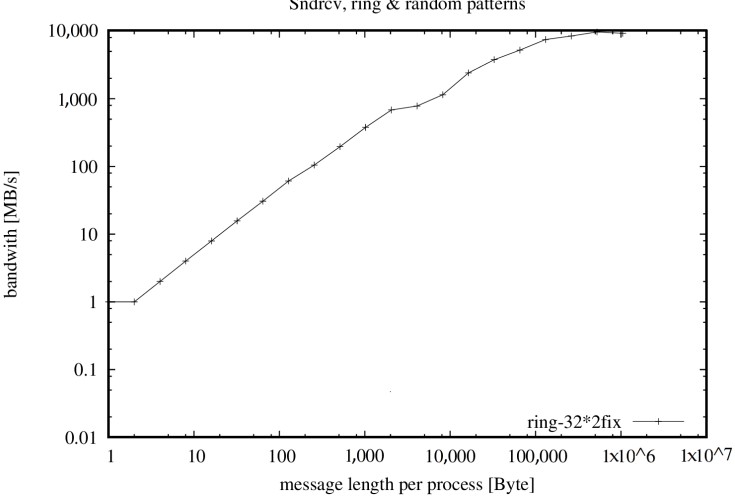

**Figure 5.** Bandwidth with 64 cores, two processes per node with span [ptile = 2] option added on queue system.

## 3. Results and Discussion

Note: All tests were performed in the warranty configurations and under the supervision of the administrator.

### 3.1. Linpack Results

Two different linpacks were run to measure the performance of CPUs and GPUs independently.

### 3.1.1. HPL with CPUs

In this test, we counted the CPU cores of the Power 8 to run the tasks and obtain the performance. Due to administrative constraints in the "Quinde I" supercomputer, we were only able to run the HPL tests on a few nodes and obtained the following results:

**HPL benchmark on "Quinde I" (only CPU's).**

Figure 6 shows that the performance is directly proportional to the input value **N** (HPL parameter specified in Section 2.2). In this test, it achieved 43.3 Gflops with a **N** of 79,296.

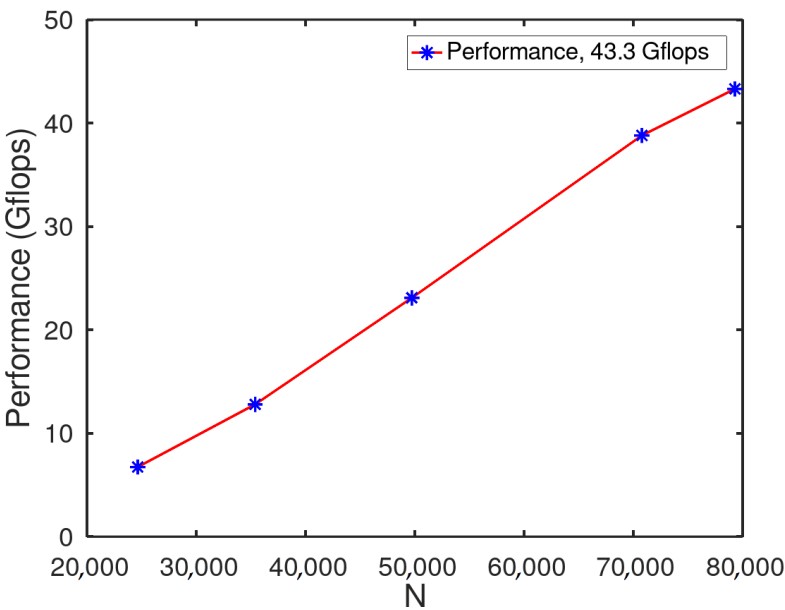

**Figure 6.** N vs. Gflops.

**HPL benchmark on 'Quinde I' ( CPU's only).**

As shown before in Figure 7, the performance grew directly proportional to the number of cores. In this test, the maximum performance was not achieved due to limitations in accessing supercomputer nodes. The maximum number of cores used in the test was 40, which was equivalent to 25% of a single node of "Quinde I". However, we can see in Figure 7 that the slope in the last interval (35–40 cores) showed a remarkable decrease. This indicates that the maximum power was reached when the number of cores, **N**, reached 40. HPL with GPUs To measure the performance of the GPUs installed on 'Quinde I' (NVIDIA TESLA K80), we needed to compile the linpack using the CUDA libraries that helped in using the GPUs. The two most important parameters to run this benchmark were:

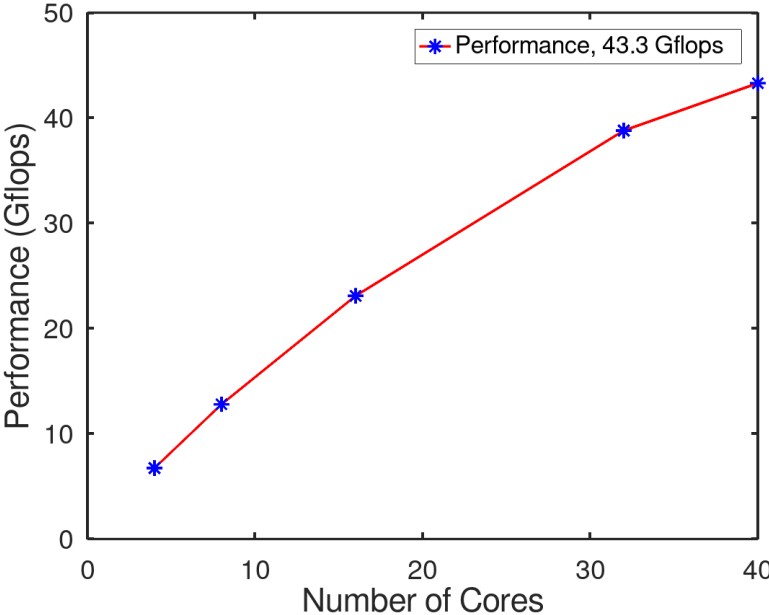

**Figure 7.** Number of cores vs. gflops.

1.  **MP PROCS = 4**
2.  **CPU CORES PER GPU = 5**

This test was performed on six nodes and the number of cores, **N** of **103,040**. The results obtained after the test were:

**HPL benchmark in 'Quinde I' (with GPU's).**

In Figure 8, the result was obtained uniformly over time. It showed how the performance grew to reach its best point which was 3632 Gflops, the main change was in the first 50 s which was due to the CPU transfer of information in the GPUs. The GPU test gave a much better performance compared to CPUs. "Quinde I" achieved 231.9 teraflops in its official benchmark, which seems reasonable due to the results obtained at only six nodes. However, according to the literature, each GPU (four GPUs) showed a performance of 2.91 teraflops [25], but it was achieved at its maximum frequency and an optimized input file was used to consider the above parameters.

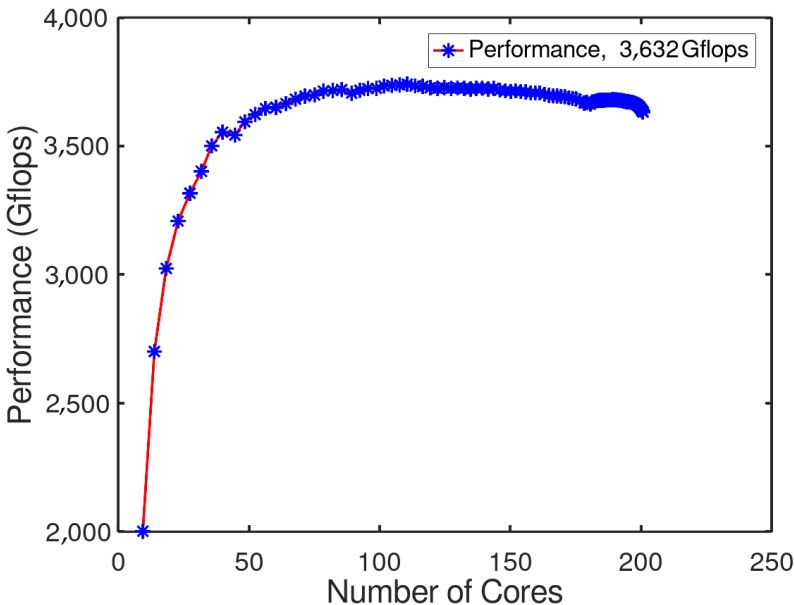

**Figure 8.** Power vs. time.

3.1.2. HPL Analysis

In order to compare and understand the results obtained, it is necessary to know that CPU's (central processing units) and GPU's (graphic processing units) are almost the same thing which have an integrated circuit with many transistors that perform mathematical calculations. The main difference is that the CPU is a general purpose processor that can perform any type of calculation, while the GPU is a specialized processor that optimizes work with large amounts of data and performs the same operations. In addition, although they are dedicated to computing, both have a significantly different design. The CPU is designed for serial processing: it consists of a few very complex cores that can run a few programs at a time. In contrast, the GPU has hundreds or thousands of simple cores that can run hundreds or thousands of specific programs simultaneously. The tasks that the GPU handles require a high degree of parallelism and therefore CUDA is needed. A GPU with its thousands of cores working in parallel can increase the performance of a CPU many times over for operations, such as a computation that requires large vector and matrix operations. Therefore, adding GPUs to the Power8 processors from the "Quinde I" can significantly increase performance. For example, in [26], the authors proved that a hybrid implementation (CPUs and GPUs) of a graph algorithm (BFS) provides the best performance over only multicore or GPUs. In this work, the performance achieved by

CPUs was lower than GPUs and it was possible to claim the large gap between the results. However, the maximum performance of CPUs and GPUs could not be exploited as they were set to moderate performance to maintain the guarantee. Additionally, in the HPL performed on GPUs, it was not possible to see how the performance changes when the **N** value was varied, since the GPU test was provided by the manager and the **N** value was already fixed.

*3.2. Stream Benchmark*

To observe the impact on performance, two different numbers of threads were used for parallel regions (shown in Tables 4 and 5). After running the test, the following results were obtained:

**Table 4.** Streaming results with OMP NUM THREADS = 10.

| Function | Best Rate MB/s | Avg Time | Min Time | Max Time |
|----------|----------------|----------|----------|----------|
| Copy: | 67,630.9 | 0.127412 | 0.127018 | 0.127911 |
| Scale: | 70,672.1 | 0.121959 | 0.121552 | 0.122957 |
| Add: | 81,089.9 | 0.159328 | 0.158904 | 0.159597 |
| Triad: | 81,356.5 | 0.158727 | 0.158383 | 0.159373 |

**Table 5.** Streaming results with OMP NUM THREADS = 20.

| Function | Best Rate MB/s | Avg Time | Min Time | Max Time |
|----------|----------------|----------|----------|----------|
| Copy: | 135,680.4 | 0.064878 | 0.063313 | 0.069888 |
| Scale: | 140,497.6 | 0.065011 | 0.061142 | 0.073112 |
| Add: | 160,384.9 | 0.083582 | 0.080341 | 0.091937 |
| Triad: | 160,827.3 | 0.085327 | 0.080120 | 0.095353 |

As expected, the bandwidth performance was better during the test when the number of threads = 20. However, if the number of threads increased, the performance got an opposite effect.

In order to analyze and compare the results obtained, Figure 9 the histogram analysis of our test case results was verified with previously shown publicly available tests on (i) Power 750 systems and (ii) a similar system using Power 8 as reference. The IBM Power 750 had four sockets containing a total of 24 cores with a clock speed of 3.22 GHz. The characteristics of the IBM Power 750 were similar to, but lower than, the Power 8 processors used by "Quinde I" used. Looking at the histogram, we can see that the characteristics of IBM Power 750 were similar but lower in operational performance (Copy, Scale, ADD, Triad) than our case study system "Quinde I (NVIDIA K80 )". We also observed that the bandwidth performance of "Quinde I (NVIDIA K80)" was higher than IBM Power 750 in all operations of the stream benchmark. The observed difference between the achieved performances represented a 20% from the maximum performance achieved by IBM power 750. This result looked reasonable due to the improvements in the characteristics among them. According to the results obtained by [27] in a Power S822LC processor, in Figure 9, "Quinde I" did not achieve its best performance. Although the compilation was the same in both cases, we assume that the difference probably came from the issue of preset configurations in "Quinde I". In the reference case, the author set the maximum frequency CPU. In our case, it could not be set because these operations jeopardize the system guarantee. In Figure 9, however, we see that the performance of Quinde I was comparable to that of the Power S822LC processor, with subtle changes in its operating dimensions.

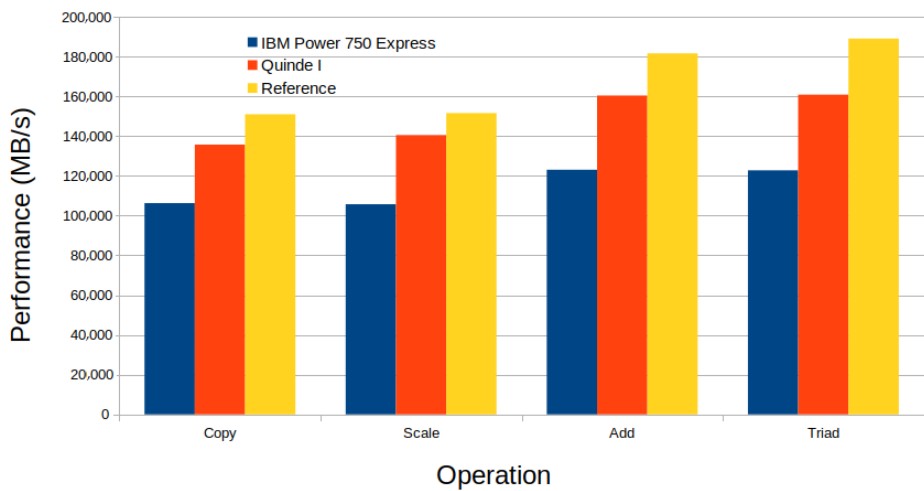

**Figure 9.** Histogram analysis of 'Quinde I Vs IBM Power 750 and Power S822LC processor for reference.

*3.3. Babel Stream Results*

"Quinde I" had NVIDIA **K80** as its graphics card. To analyze the performance, the test was run several times with the original card configurations (clock frequency = 640 MHz) and varying the input array size, obtaining the following results:

**GPU 1 Stream Benchmark**

Figure 10 shows the behavior for the five operations performed by the benchmark at 640 MHz. When the array size reached $5 \times 10^6$, the GPU bandwidth reached the maximum speed of 170,000 MB/s. However, a look at the literature of the previously published paper [28] shows that the expected performance achieved by Babel Stream Benchmark on an NVIDIA K80 tesla was 176,000 MB/s. We confirmed that for this reason, the GPU clock speed was changed to achieve the maximum performance of the graphics card.

**Note: The test was approved and supervised by the manager of "Quinde I".** The test was run with an array size of $3.5 \times 10^{10}$ and the following changes in the graphics card configurations:

- nvidia-smi-application-clocks = memory clock speed, clock speed
- nvidia-smi-application-clocks = 2505 MHz, 705 MHz
- nvidia-smi-application-clocks = 2505 MHz, 810 MHz
- nvidia-smi-application-clocks = 2505 MHz, 875 MHz

Plotting the obtained data, Figure 11 was obtained.

**GPU 1 Stream Benchmark**

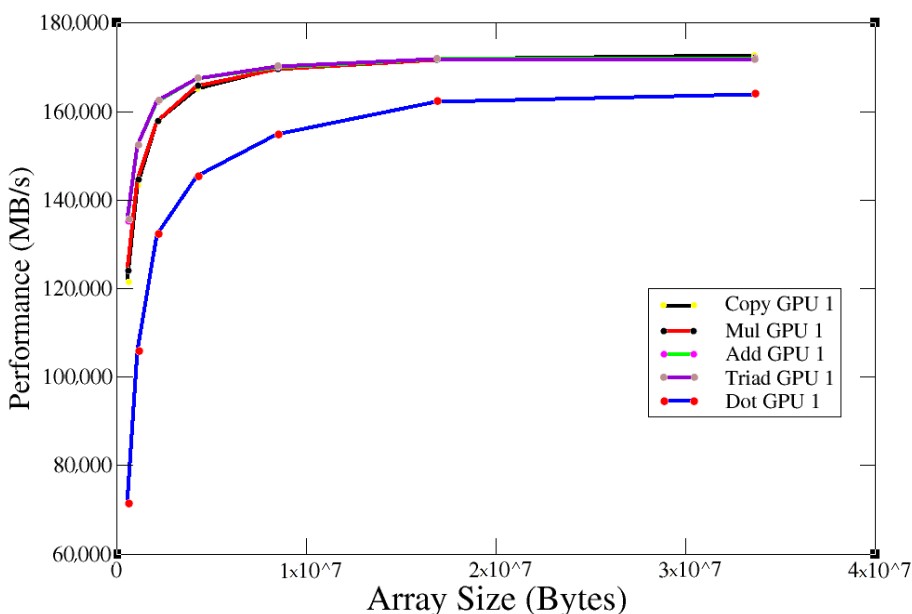

**Figure 10.** K80 performance results.

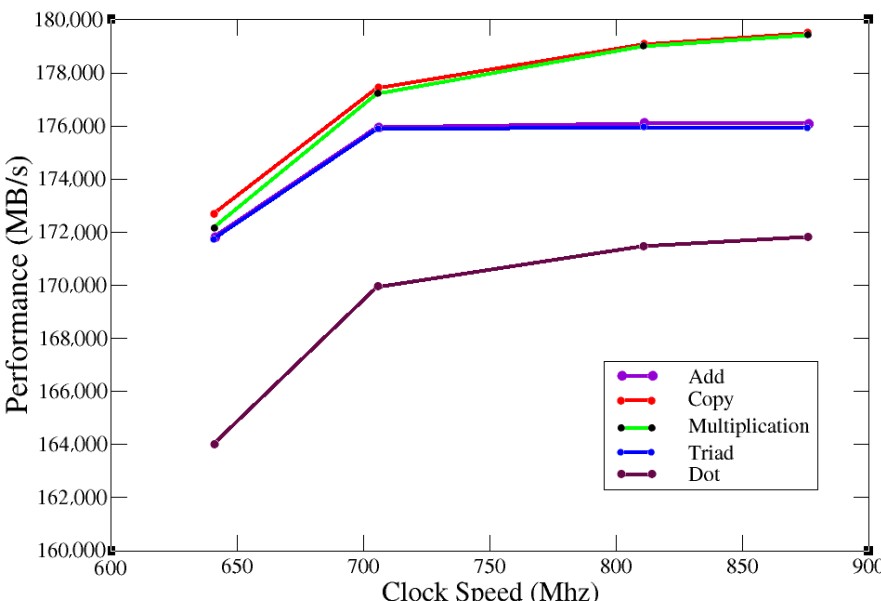

**Figure 11.** K80 performance results.

Figure 11 shows that better bandwidth performance was achieved, even outperforming the literature reference [28]. With the maximum clock rate (875 MHz), a bandwidth of 180,000 MB/s was achieved. The obtained results are attributed to the optimization of the compilation phase.

NVIDIA TITAN XP vs. NVIDIA K80

To observe and compare the performance of the NVIDIA K80 graphics card, the results from the test on an NVIDIA TITAN XP (latest model) and the following results were obtained:

In Figure 12 The bandwidth performance of a TITAN XP was significantly higher than that of a K80. The bandwidth achieved was more than twice the K80 bandwidth. This comparison gave an idea of how fast the technology was evolving in both processor speed

and bandwidth. On the other hand, the analysis of the performance of the two GPUs based on their theoretical performance that we obtained is shown in Figure 13. In theory, NVIDIA k80 was much better than NVIDIA TITAN, but in the test Figure 12 performed, exactly the opposite happened.

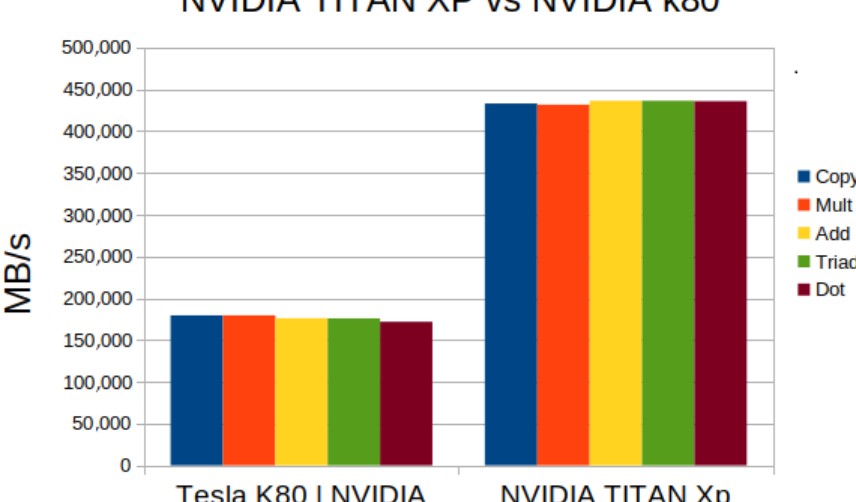

**Figure 12.** NVIDIA K80 (2014) vs. NVIDIA TITAN XP (2017).

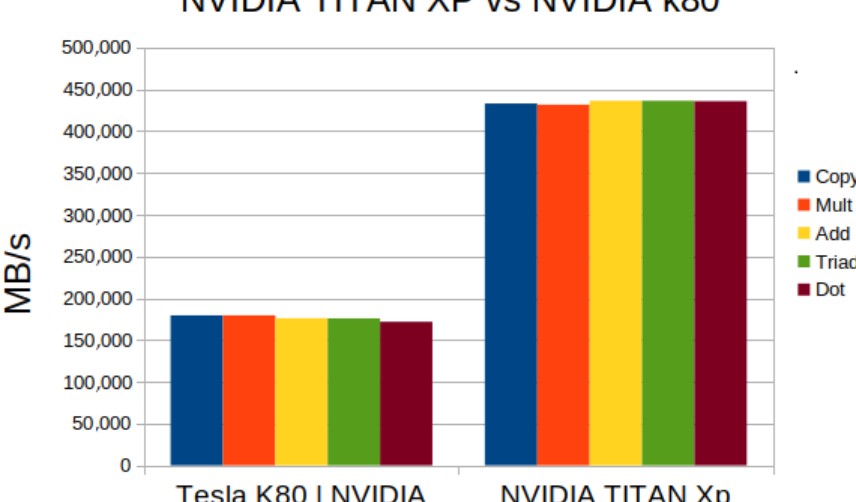

**Figure 13.** NVIDIA K80 vs. NVIDIA TITAN xp.

Another measure to compare the performance of CPUs and GPUs is to calculate the balance. In this work, the balance was calculated for both GPUs:

$$\text{Balance} = \frac{\text{sustainable-memory-bandwidth (Byte/s)}}{\text{Performance (Flops/s)}} \tag{2}$$

- **NVIDIA K80: 0.137855579868709**
- **NVIDIA TITAN xp: 1.16064362964917**

In balance, when the system is close to 1, it indicates excellent balance, and far at 1 it behaves just the opposite. The NVIDIA TITAN xp was close to 1, so it was better balanced than the NVIDIA K80. Improving memory bandwidth instead of CPU speed resulte in better performance, even if the CPU speed was lower. In other words, the performance of an NVIDIA K80 was wasted because it processed data very quickly and had to wait for data to be written or read from memory.

### 3.4. DEGMM Results

After performing the test on each node of 'Quinde I', the results are described in Figure 14. DGEMM Individual Performance:

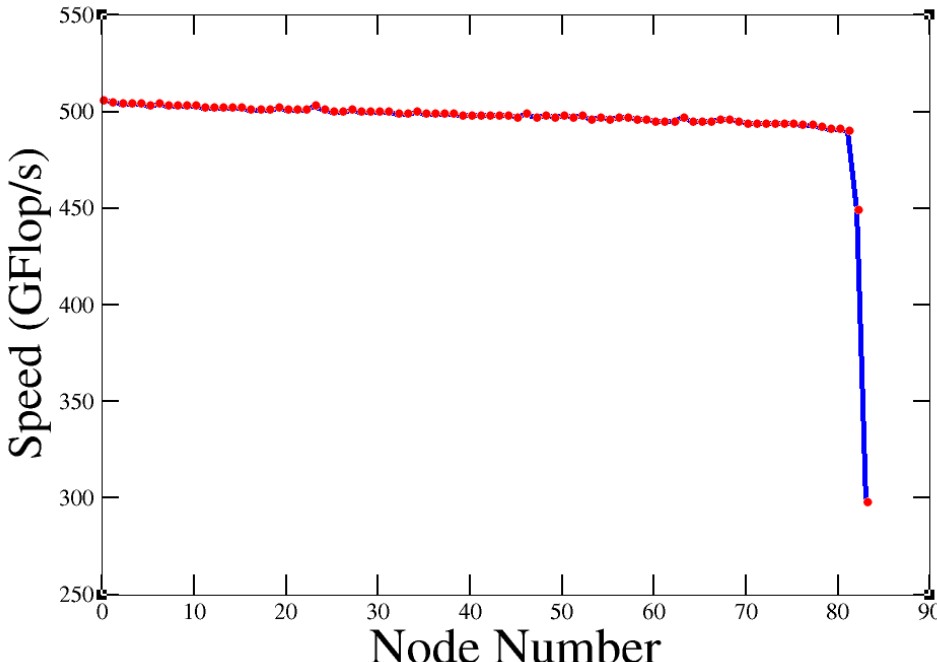

**Figure 14.** Speed from each node of "Quinde I".

Almost all (except two) have a similar rate (close to 500 Gflop) for floating point operations. The value obtained in the test agrees with the theoretical performance of each node: 504 Gflops. Which comes from:

$$\text{Flops} = \text{Sockets} \cdot \frac{\text{Cores}}{\text{Socket}} \cdot \frac{\text{Cycles}}{\text{Second}} \cdot \frac{\text{Flops}}{\text{Cycle}} \tag{3}$$

where: Sockets = 20, Cores/Socket = 8, Cycle/Second = 3.5 GHz, Flops/Cycle = 0.9.

Figure 14 shows abnormal performance. This could be an indication that there was something wrong with these nodes, so they should be checked. After the test, the nodes were checked and the problems were fixed. If the nodes were performing normally, we got Figure 15.

**DGEMM Individual Performance**

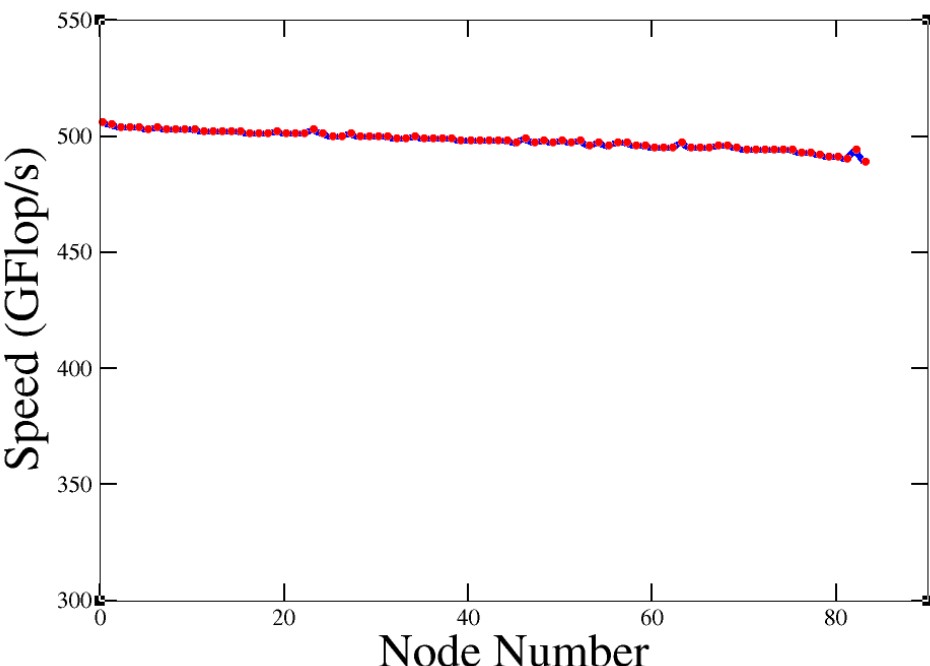

**Figure 15.** Performance of the nodes after the check.

*3.5. LLCBench—Cache Bench Results*

Cache-Bench on a Node of "Quinde I"

Figure 16 shows the eight operations and results produced by the test on a single node of 'Quinde I'. The maximum vector used in the test was smaller than the cache memory, which was evident in the performance as the cache read was constant throughout the test, which means that the data were obtained directly from the cache. The cache write varied during the test, but it held the interval of 2000 MB/s. This could be because this benchmark was heavily influenced by architectural quirks in the memory subsystem such as replacement policy, associativity, blocking, and write buffering. The cache RMW generated twice as much memory traffic because each data item must first be read from memory/cache to register and then back to cache. The bottom line is that the test showed that the performance of was greater than the addition of read and write performance. In terms of tuned versions, the Cache Read test showed better performance. This means that the compiler was not doing a good job on this type of simple loops. However, the tuned cache RMW showed worse performance, which means that the compiler did a better job than the tuned version. Finally, for cache writing, the behavior of the normal and tuned versions was too similar, which also indicates that the compiler was doing a good job. Finally, the libraries provided by C showed better performance than the routines used by the compiler, especially memcpy, which reached the best performance point of almost 90,000 MB/s. It is attributed the compiler used in this work, the smpi, a special and dedicated compiler to IBM.

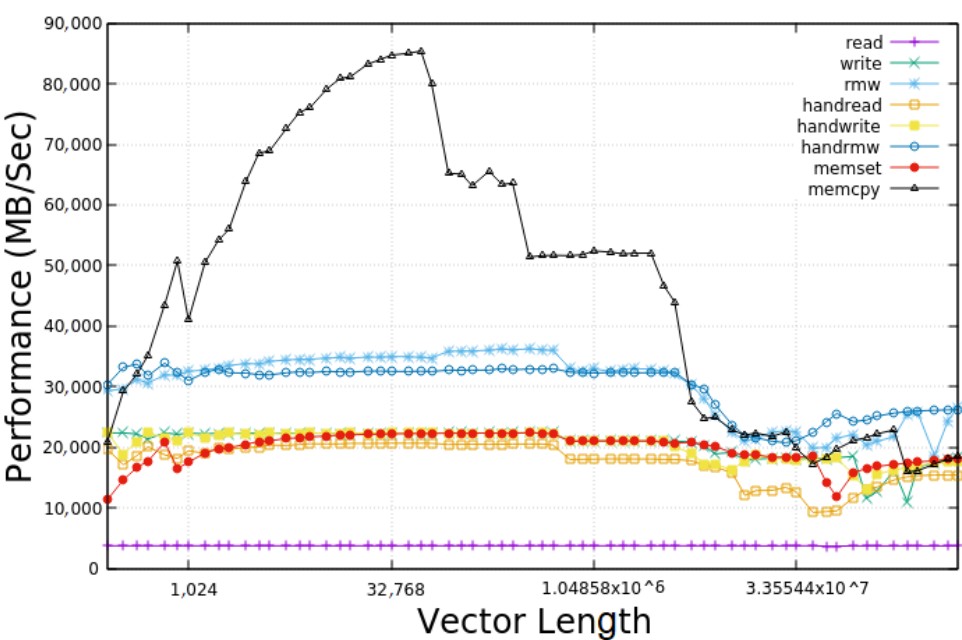

**Figure 16.** Cache Bench results of Quinde I.

### 4. SRUMMA vs. DGEMM Algorithm on "Quinde I"

In this section, we address a common problem in various domains, namely, obtaining the effective product of C = AB, where A and B are large and dense N × N matrices. In this study, we compared and evaluated the performance of two main algorithms: SRUMMA and DGEMM. SRUMMA algorithm [15] is a parallel algorithm that implements serial block-based matrix multiplication with algorithmic efficiency. It is suitable for clusters and scalable in shared memory systems. SRUMMA differs from other parallel matrix multiplication algorithms by explicitly using shared memory and remote memory access (RMA) communication instead of message passing, the usual process communication protocol. As a result, the algorithm is completely asynchronous and avoids any processor coordination that arises from this protocol. In [15], the authors specified the parallel time of the algortihm as follows:

$$T_{par\_rma} = O(\frac{N^3}{P}) + O(\sqrt{P}) \tag{4}$$

where: *P* = number of processors. *N* = size of the matrix.

Second, DEGMM is a double-precision matrix-matrix multiplication defined in [29] as:

$$C = \alpha \cdot A \cdot B + \beta \cdot C \tag{5}$$

where *A*, *B* and *C* are matrices and *α* and *β* are scalars. In this work, we assume $\alpha = 1$ and $\beta = 0$. DGEMM is implemented as a subroutine of LAPACK, which provides routines for solving systems of simultaneous linear equations, least-squares solutions of systems of linear equations, eigenvalue problems, and singular value problems. A simple, straightforward implementation of DEGMM consists of three nested loops. The DGEMM algorithm is a well-known blocking algorithm [29] where the computation is done on a two-dimensional grid of threaded blocks [30] and here lies the main difference between the algorithms. We compared the performance of the algorithms with different number of cores and the same N (matrix dimension). SRUMMA vs DGEMM algorithm.

In the experimental results, we can observe in Figure 17 that the SRUMA algorithm with an N of 8192 achieved 273 Gflops with 64 cores, but when the number of cores

increased, the efficiency of the algorithms decreased. On the other hand, we can see that DGEMM showed poor performance compared to SRUMMA. We notice that it became constant when we increased the number of cores, and we can assure that SRUMMA algorithm gave better performance than the most popular algorithms used today. However, we can state that it did not have good scalability, since its performance decreased very quickly after a certain point. Moreover, when the value of **N** in SRUMMA was very large, the algorithm interferes with the load sharing facility (LSF) service of "Quinde I", so we could not increase the value of **N**.

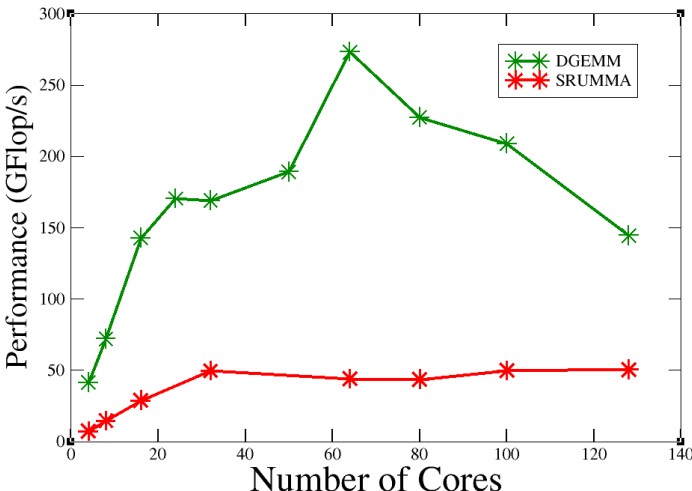

**Figure 17.** Performance of SRUMMA vs. DGEMM.

### 5. Conclusions

In conclusion, in this paper it has been shown that "Quinde I" has a correct performance in all measured aspects according to the literature. However, in Figure 9, two nodes show abnormal performance, which was checked and later corrected by the manager. When the SRUMMA algorithm was applied, a significant performance improvement was observed over the conventional algorithms such as DEGMM used in this work. Many tests carried out in this work were limited by the administrative and warranty conditions of the "Quinde I" by the supplier, therefore the maximum performance was not achieved. However, the performance achieved during the tests was still strong. It can be clearly seen that the options (copy, scale, add, triad) that slightly surpass the previous model IBM Power 750 Express (as shown in Figure 9), do not reach the reference due to the configuration mentioned above in the explanation, however, a clear technical definition can be given about exceeding the manufacturer's reference as a supervised technical test was performed and this can be seen in Figure 11, it can be clearly seen that when the clock frequency is matched to that of the manufacturer, all the calculated measurements far exceed the reference in the literature, which confirms the computational performance. Additionally, when performing work with the use of NVIDIA K80 Tesla graphics cards, with reference to NVIDIA Titan XP, the configuration given in the explanation is visualized in Figure 12, shows that the performance is lower than that of NVIDIA Titan, however, by performing a special supervised test and changing the speeds of the graphics card, as shown in Figure 10, that it exceeds in large quantity even on the supplier reference, so it can be concluded that when the size of the array is up to $5 \times 10^6$, the GPU bandwidth reaches the maximum speed, which is about 170.000 MB/s. However, a look at the literature shows that the expected performance achieved by Babel Stream Benchmark on an NVIDIA K80 tesla is 176,000 MB/s [27]. For this reason, the clock frequency of the GPU was changed in a monitored manner to obtain the maximum performance of the GPU and confirm that it outperforms the reference, as can be seen in Figure 11.

**Author Contributions:** Methodology, F.R.J.P., J.C.L.L.; software, G.E.C.C.; validation, G.E.C.C.; formal analysis, E.P.E.R., G.E.C.C.; investigation, E.P.E.R., G.E.C.C., F.R.J.P. and S.P.T.; resources, G.E.C.C.; data curation, F.R.J.P.; project administration, F.R.J.P. All authors have read and agreed to the published version of the manuscript.

**Funding:** This research was funded by the seed grant "Computational modelling of biomaterials and applications to bioengineering and infectious disease, Universidad Technologica Indoamérica, Ecuador" awarded to S.P.T.

**Institutional Review Board Statement:** Not applicable.

**Informed Consent Statement:** Not applicable.

**Data Availability Statement:** Not applicable.

**Acknowledgments:** Giovanny Caluña thank Quinde-I super computing facility of Siembra EP, Urcuqui, Ecuador, to conduct internship opportunity, Likewise, thank Alexandra Nataly Culqui Medina for her contributions in networking and being able to carry out this work.

**Conflicts of Interest:** The authors declare no conflict of interest.

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
