# Peer review of "Dense Matrix Multiplication Algorithms and Performance Evaluation of HPCC in 81 Nodes IBM Power 8 Architecture"

_computation, doi:10.3390/computation9080086_

Round 1

Reviewer 1 Report

Title: Dense matrix multiplication Algorithms and Performance Evaluation of HPCC in 81 nodes IBM power 8 architecture
In the paper the authors performs a wide range of benchmarks on an HPC system. They run several benchmarks that cover the main aspects that impact performance on HPC systems. I have some comments that are listed below:

- In fig. 7 and 8 I would add a reference performance value (i.e. the theoretical peak performance or a fraction of that) directly in the plot in order to have a visual comparison with the measured one;
- Since the dense matrix multiplication algorithms is part of the title, I suggest to add more details on the implementation of those algorithms.
- Line 255 do you mean Number of Cores instead of N?
- There are several missing spaces/capital letters after the dots, I list some: line 40 (rankings.A) , 48 (performance.considering), 59 (forecasting. one), 310 (Also, We)
- At line 141,142 both  "Benchmark Methods" and "HPL" appear on the same subsection level.
- I suggest to write explicitly in the text the definition and the importance of the sustained floating-point rate Flops/s.
- Line 443 textbf 5times 10^6

Reviewer 2 Report

This paper shows "Quinde I" presents a correct performance according the literature in all the aspects which were measured. The topic is interesting. The paper can be improved following the suggestions below.

1. In the introdcution, the paragraph starting with Formally, benchmark is a program or a set of programs which are run on a single machine or cluster in order to achieve the maximum performance of a certain feature under specific conditions, and be able to compare the performance results with the measures obtained of similar equipment or the expected values obtained in theory' is not very easy to read. Maybe some examples would be useful.
2. Is adding graphics processing units to server hardware is a good way to increase computer power? Some explanations and comparision are needed. 
3. Some figures are not of good quality. For example, Figure 2 is blurred. Please use high quality figures. 
4. Although a small NB improves the load balance, but the change is very minor. Could you comment on other alternatives?
5. Algorithm 1 would benefit from a complexity analysis. 
6. Please double check the details in The IBM Power 750 has four 3.3 GHz POWER7 processors with a total of 24 cores, 128 GB (32 x 4 GB) DDR3 1066MHz, 32KBI+32KBD L1 on chip per core, 256KB L2 on chip per core, and 4MB L3 on chip per core. 
7. Equations (4)-(7) are written using strange fonts. This is enigmatic. Please carefully check throughout the paper to make sure all expressions are consistent.

Round 2

Reviewer 2 Report

The paper has been improved with necessary amendment. I am happy to recommend it for publication.